# Assessment of missed opportunities for vaccination in Kenyan health facilities, 2016

**Anyie J. Li**[1,2]*, **Collins Tabu**[3], **Stephanie Shendale**[4], **Kibet Sergon**[5], **Peter O. Okoth**[6], **Isaac K. Mugoya**[7], **Zorodzai Machekanyanga**[8], **Iheoma U. Onuekwusi**[5], **Colin Sanderson**[9], **Ikechukwu Udo Ogbuanu**[2,4]

**1** ASPPH/CDC Allan Rosenfield Global Health Fellowship and PHI/CDC Global Health Fellowship, Atlanta, GA, United States of America, **2** Global Immunization Division, Centers for Disease Control and Prevention, Atlanta, GA, United States of America, **3** National Vaccines and Immunization Program, Ministry of Health Kenya, Nairobi, Kenya, **4** Department of Immunization, Vaccines and Biologicals, World Health Organization, Headquarters, Geneva, Switzerland, **5** World Health Organization Kenya, Country Office, Nairobi, Kenya, **6** UNICEF Kenya, Country Office, Nairobi, Kenya, **7** Maternal and Child Survival Program, Nairobi, Kenya, **8** Inter-Country Support Team (IST)–East and Southern Africa, World Health Organization, Harare, Zimbabwe, **9** London School of Hygiene and Tropical Medicine, London, United Kingdom

\* anyieli@cdc.gov

**Data Availability Statement:** All relevant data are within the manuscript.

**Funding:** AJL is supported by Cooperative Agreement Number U36OE000002 from the Centers for Disease Control and Prevention (CDC)

## Abstract

### Background

In November 2016, the Kenya National Vaccines and Immunization Programme conducted an assessment of missed opportunities for vaccination (MOV) using the World Health Organization (WHO) MOV methodology. A MOV includes any contact with health services during which an eligible individual does not receive all the vaccine doses for which he or she is eligible.

### Methods

The MOV assessment in Kenya was conducted in 10 geographically diverse counties, comprising exit interviews with caregivers and knowledge, attitudes, and practices (KAP) surveys with health workers. On the survey dates, which covered a 4-day period in November 2016, all health workers and caregivers visiting the selected health facilities with children <24 months of age were eligible to participate. Health facilities (n = 4 per county) were purposively selected by size, location, ownership, and performance. We calculated the proportion of MOV among children eligible for vaccination and with documented vaccination histories (i.e., from a home-based record or health facility register), and stratified MOV by age and reason for visit. Timeliness of vaccine doses was also calculated.

### Results

We conducted 677 age-eligible children exit interviews and 376 health worker KAP surveys. Of the 558 children with documented vaccination histories, 33% were visiting the health facility for a vaccination visit and 67% were for other reasons. A MOV was seen in 75% (244/324) of children eligible for vaccination with documented vaccination histories, with 57% (186/324) receiving no vaccinations. This included 55% of children visiting for a

and the Association of Schools and Programs of Public Health (ASPPH) (https://www.aspph.org/study/fellowships-and-internships/) and NU2GGH002093-01-00 from the CDC and the Public Health Institute (PHI) (https://phi-cdcfellows.org/). The funders had no role in study design, data collection and analysis, decision to publish, or preparation of the manuscript.

**Competing interests:** The authors have declared that no competing interests exist.

vaccination visit and 93% visiting for non-vaccination visits. Timeliness for multi-dose vaccine series doses decreased with subsequent doses. Among health workers, 25% (74/291) were unable to correctly identify the national vaccination schedule for vaccines administered during the first year of life. Among health workers who reported administering vaccines as part of their daily work, 39% (55/142) reported that they did not always have the materials they needed for patients seeking immunization services, such as vaccines, syringes, and vaccination recording documents.

## Conclusions

The MOV assessment in Kenya highlighted areas of improvement that could reduce MOV. The results suggest several interventions including standardizing health worker practices, implementing an orientation package for all health workers, and developing a stock management module to reduce stock-outs of vaccines and vaccination-related supplies. To improve vaccination coverage and equity in all counties in Kenya, interventions to reduce MOV should be considered as part of an overall immunization service improvement plan.

## Background

A missed opportunity for vaccination (MOV) includes any contact with health services by a child (or adult) who is eligible for vaccination (unvaccinated, partially vaccinated, or not up-to-date, and free of contraindications to vaccination), which does not result in the individual receiving all the vaccine doses for which he or she is eligible [1, 2]. Studies have shown that MOV can occur for a variety of reasons including health workers not checking vaccination status, limited integration of vaccination services with other health services, a shortage of staff administering vaccines, poor vaccination card retention, and stock-outs of vaccines or related supplies [1, 3–9]. MOV may be hindering countries from increasing their vaccination coverage; successful efforts to address MOV have the potential to help countries reach their immunization targets, improve timeliness, and promote integration between health programs.

Globally, the first systematic literature review of MOV identified a global median MOV prevalence of 32% among both children and women of childbearing age who visited a health center and 67% among the subpopulation of women and children eligible for vaccination at the time of visit [1]. An updated systematic review published in 2014 identified the same global median MOV prevalence of 32% among children and 47% among women of childbearing age who visited a health center [10]. Unfortunately, these systematic literature reviews of MOV prevalence have shown limited progress in the reduction of MOVs globally over the course of the past 20 years [1, 10].

In 2008, Kenya endorsed a national multi-year strategic plan for development, *Vision 2030*, which has set a target of 90% vaccination coverage for all infants of all recommended vaccines [11]. Since then, the Kenya National Vaccines and Immunization Programme (NVIP) has introduced several new vaccines including pneumococcal conjugate vaccine (PCV) (2011), the second dose of measles-containing vaccine (MCV) (2013), and rotavirus vaccine (2014) [12, 13]. In 2016, however, 35% of the annual birth cohort remained under-vaccinated [14, 15]. A proportion of these children in Kenya may already be accessing health facilities for other health services but may be missed for vaccination. A review of Demographic and Health Survey data in 2014 found an MOV prevalence of 42% and a study of children of Maasai nomadic

pastoralists conducted in 2016 found the prevalence of MOV to be 30% [4, 16]. Another study among children in an urban poor settlement of Nairobi, Kenya found that 22% of children who were fully immunized by 12 months had received their vaccine doses out of the recommended order [17]. Unfortunately, previous studies assessing MOV or factors related to MOV have been limited in scope and have used varying methodologies, leading to limitations in comparability [4, 16, 18–23].

In 2016, the Strategic Advisory Group of Experts (SAGE) on Immunization endorsed the updated World Health Organization's (WHO) strategy for assessing MOV [24]. The standardized methodology has three phases: a desk review, facility-based field assessment of MOV (primary data collection including quantitative and qualitative field work), and an intervention phase. The updated methodology focuses on triangulation of data to develop actionable country-driven interventions to reduce MOV [2, 25, 26]. As of April 2019, a total of 12 countries across four WHO regions have implemented this methodology [7–9, 27, 28]. Recent studies in Chad, Malawi and Timor Leste using the updated WHO methodology showed that between 41–66% of eligible children had made contact with the health system and were not vaccinated with all the vaccines for which they were eligible [8, 29]. In November 2016, to better understand the reasons for under-vaccination and to prioritize needed interventions, the Kenya NVIP conducted a MOV assessment using the updated WHO methodology [28]. This paper details the findings from the quantitative component of the MOV assessment. The results of the qualitative component of the Kenya MOV assessment have been described in detail in a separate manuscript [28].

## Methods

### Study design

A cross-sectional study design employing both qualitative and quantitative methods was used. Study aims included understanding how many MOVs are occurring, why they are occurring, and what interventions can be implemented to address identified gaps and barriers to full vaccination of infants and children [2]. The quantitative component of the WHO MOV methodology, described in this paper, included exit interviews with caregivers of children <24 months and anonymous self-administered health worker knowledge, attitude and practices (KAP) surveys. The WHO guides on the MOV strategy and past MOV assessments provide more detailed information on the general process and expected outcomes [2, 7, 8, 25–28].

### Data collection tools

Prior to deployment for field work, field staff adapted the data collection tools (caregiver exit questionnaire and the health worker KAP) to the Kenyan context from the generic questionnaires available from WHO [2, 26]. The caregiver exit questionnaire collected demographic information, vaccination history, awareness of routine immunization services, and perceived quality of the vaccination services. The health worker KAP questionnaire collected demographic information, knowledge and attitudes toward vaccination, and additional questions on vaccination practices and decision-making specifically for health workers who reported that they routinely administer vaccines as part of their daily duties. Both questionnaires included core questions and additional questions (not required), and single and multi-select responses. All questionnaires were written in English, but exit interviews were conducted in English or Swahili depending on the preference of the respondent. Health worker KAP surveys were self-administered with a surveyor available to help assist with language and the electronic platform, as needed. Prior to data collection, all field tools were pretested for country context and ease of administration [2, 26].

## Data collection

Field work for the MOV assessment took place over 10 days in November 2016. Data collectors were trained during the first three days, followed by four days of data collection and three days of data analysis, brainstorming and debrief. All data collectors reconvened for the brainstorming sessions and finalization of the intervention action plan. Data collectors consisted of staff from the NVIP, Kenya Ministry of Health (MoH), and various in-country immunization partners as well as international development partners (US Centers for Disease Control and Prevention (CDC) and WHO).

Using the guidance provided in the MOV methodology, the NVIP selected 10 counties for field data collection [26]. The counties selected were Bungoma, Kajiado, Kiambu, Kitui, Migori, Mombasa, Nakuru, Taita Taveta, Trans Nzoia, and West Pokot. The 10 counties represented a geographical spread across the country and various immunization performance levels (as indicated by coverage of the third dose of diphtheria-tetanus-pertussis-hepatitis B-*Haemophilus influenzae* type b or *pentavalent* vaccine). In each county, from a list of all health facilities, the MOV strategy team purposively selected four health facilities for quantitative data collection, regardless of whether or not they were providing immunization services daily. The health facilities included were of varying sizes (Kenya Essential Health Package [KEPH] levels 2–5), ownerships (MoH, Non-Governmental Organization [NGO], faith-based, private), and types (hospitals, health centers, and dispensaries). Due to the limited timeframe for data collection, the MOV strategy team also took logistics and ease of accessibility into account when determining the final sample of health facilities.

Teams of three to four data collectors were deployed to each of the 10 selected counties. Team members were purposively assigned by the MOV strategy team to ensure a mix of representation from local immunization partners, gender, local language ability, and field survey skills.

Data collectors conducted exit interviews as caregivers were leaving each health facility after receiving health services. Each team spent one day per health facility. Each team was expected to conduct 20 exit surveys per health facility per day, approaching sequential caregivers leaving the health facility until they achieved 20 completed exit surveys. All caregivers who were 15 years of age or older and who were leaving the health facility with a child <24 months on the days of field work were eligible and were requested to participate in the survey. If a caregiver was accompanied by more than one child, the survey questions were asked about the youngest child. No specific efforts were made to obtain equal samples of children in different age groups (e.g. <12 months or between 12 and 24 months). Data collectors recorded the child's dates of vaccination from their mother-and-child health (MCH) booklets, which contains a child's comprehensive vaccination history in Kenya, or other temporary documents. If mothers did not have documentation of the child's vaccination dates, data collectors requested basic demographic information for use to later abstract vaccination dates from the health facility register following the survey. Oral vaccination histories were not accepted as a substitute. If no MCH booklets or temporary document was available, and the data collector was unable to locate the child's records in the health facility register, no vaccination dates were recorded in the questionnaire.

For the health worker KAP surveys, all health workers at the health facility, regardless of whether routine immunization service delivery was part of their daily work, were requested to participate. Each team was expected to conduct 10 health worker KAP surveys per health facility visited.

All data were collected electronically using a tablet survey software platform (*Zegeba AS* [Alesund, Norway]). Data collection teams were assigned unique logins for the survey

platform specific to their field site. Only key study staff from the MOV strategy team had access to all survey data. Surveys were uploaded to a secure network daily.

## Data analysis

Survey data were downloaded directly (in Excel format) from the secure electronic network for analysis. Data were analyzed using STATA (version 14.2, College Station, Texas). Following standard methodology used in the analysis of previous MOV assessments, we created a flow chart to identify children with MOV (Fig 1) [1, 8]. We created frequency distributions for children with documented vaccination dates, and those eligible for one or more vaccine doses at the visit. We calculated MOV based on the child's date of birth and interview date, the national schedule, and the presence of contraindications (as reported by the caregiver). Only children who were eligible for one or more vaccine doses at the visit, and who had a documented vaccination history or evidence of a blank MCH booklet, were included in the calculation of child-based prevalence of MOV. Each child could only be eligible for one dose of a particular vaccine; if all doses in a vaccine series were overdue, only one dose could be given at the visit and thus counted only as one child-based MOV. We differentiated between children that had received all eligible doses, some, but not all the doses, and no doses. All antigens in the Kenyan national immunization schedule were included in the calculation of MOV except for inactivated polio vaccine (IPV) and yellow fever (YF) vaccine; these antigens were newly introduced

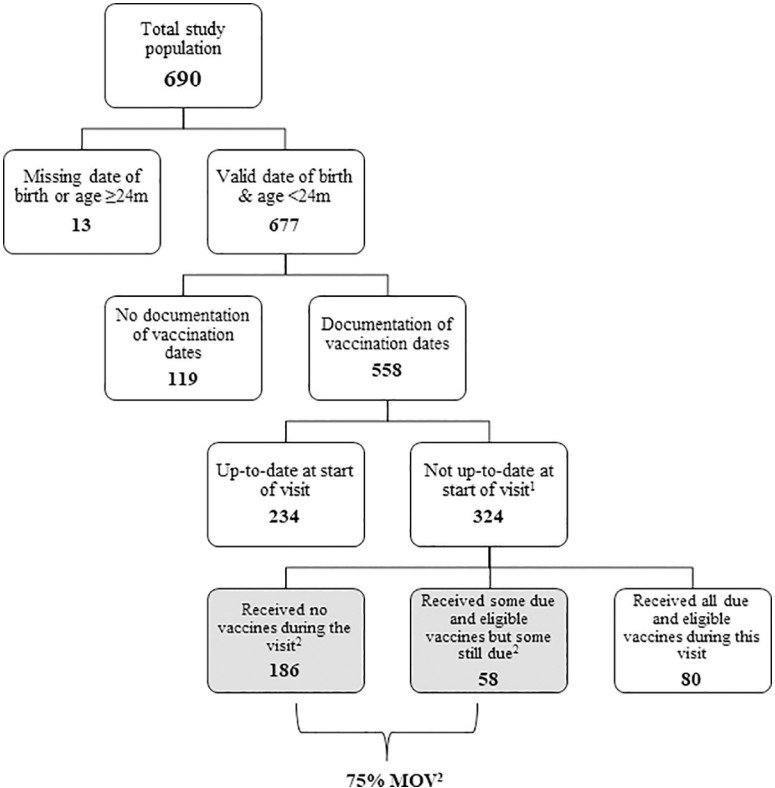

**Fig 1. Health-center-based flow-chart for determining missed opportunities for vaccination, Kenya, 2016.** [1]All children were without contraindications; [2]Missed opportunity for vaccination (MOV): contact with health services by a child (or adult) who is eligible for vaccination (unvaccinated, partially vaccinated, or not up-to-date, and free of contraindications to vaccination), which does not result in the individual receiving all vaccine doses for which they are eligible [1, 2].

and not available in every selected county at the time of the assessment. We cross tabulated MOV by key demographic variables, including reason for visit and age.

We also calculated timeliness of all vaccine doses received (not only those received on the day of exit interview). Timeliness was based on the child's date of birth, vaccination date, and the nationally recommended ages for each vaccine dose. We created timeliness intervals (too early, timely, and late) based on the national schedule and past studies (Table 1) [8, 17, 30, 31]. These categories do not imply validity of the doses given, but are simply a measure of the interval between the nationally recommended age (based on the national schedule) and the actual date of vaccination. Antigens in the Kenya NVIP have no recommended maximum interval between doses and the NVIP had lifted the age restrictions for rotavirus vaccine prior to this assessment. Since the maximum age of children in this survey was <24 months, the maximum recommended ages for all the antigens were higher than the ages of the study population [12, 32]. IPV and YF were also excluded from the timeliness analysis.

## Ethical approval

The MOV assessment protocol was reviewed by the Kenya MoH and categorized as a program assessment. As a program assessment, it was considered exempt from additional Institutional Review Board review. Prior to administering surveys, all data collectors informed participants

**Table 1. Time intervals used for classifying timeliness of vaccination doses received by surveyed children (<24 months), using the nationally recommended ages for vaccination, missed opportunities for vaccination (MOV) assessment, Kenya, 2016[1].**

| Vaccine | Recommended age of vaccination | Too early | Timely | Late |
|---|---|---|---|---|
| **Birth dose** | | | | |
| BCG[2] | Birth | -- | 0–30 days | >30 days |
| OPV[3] | | | 0–14 days | >14 days |
| **First dose** | | | | |
| Pentavalent[4] vaccine | 6 weeks (42 days) | <42 days | 42–56 days | >56 days |
| OPV[3] | | | | |
| PCV[5] | | | | |
| Rotavirus vaccine | | | | |
| **Second dose** | | | | |
| Pentavalent[4] vaccine | 10 weeks (70 days) | <70 days | 70–84 days | >84 days |
| OPV[3] | | | | |
| PCV[5] | | | | |
| Rotavirus vaccine | | | | |
| **Third dose** | | | | |
| Pentavalent[4] vaccine | 14 weeks (98 days) | <98 days | 98–112 days | >112 days |
| OPV[3] | | | | |
| PCV[5] | | | | |
| **MCV[6] first dose** | 9 months (270 days) | <270 days | 270–365 days | >365 days |
| **MCV[6] second dose** | 18 months (548 days) | <548 days | 548–730 days | >730 days |

[1] The table does not comprehensively include all vaccines listed in the national immunization schedule for children <24 months; newly introduced vaccines at the time of assessment (2016) were excluded (inactivated polio vaccine and yellow fever vaccine).

[2] bacille Calmette-Guerin (BCG) vaccine.

[3] Oral poliovirus vaccine (OPV).

[4] Diphtheria-tetanus-pertussis-hepatitis B-*Haemophilus influenzae* type b vaccine (Pentavalent).

[5] Pneumococcal conjugate vaccine (PCV).

[6] Measles-containing vaccine (MCV).

about the details of the assessment and obtained verbal consent. As there was no personally identifiable information collected, the Kenya MoH considered verbal consent to be appropriate, since participation posed minimal risk to the participants. Informed verbal consent was recorded by the interviewer on the data collection tool.

## Results

There were 690 caregivers surveyed during the exit interviews (Fig 1) and 376 health workers completed the KAP surveys. Thirteen children were excluded from the analysis of the exit surveys because of ineligibility by age or a missing date of birth. Of the remaining 677 age-eligible children, 558 children had documented vaccination dates (obtained by the data collector either from the child's MCH booklet or health facility register). Of the 558 children with documented vaccination dates, 324 children were eligible for and due for at least one vaccine during the health facility visit. None of the caregivers or health workers who were approached for inclusion refused to participate in the study.

### Caregiver exit interviews

**Demographics of children and caregivers.** Caregivers from 10 counties in Kenya were interviewed with a median of 56 interviews and interquartile range (IQR) of 33–81 interviews per county. Of 558 children with documented vaccination dates, 75% were <12 months of age and 25% were between 12 and <24 months of age (Table 2). Most interviewed caregivers were mothers (95%) and most could read and write (88%). Approximately 33% of visits were for vaccination and 67% were for other reasons.

**Vaccination status.** Among children with documented vaccination dates, 23% of caregivers (129/555) reported that the health worker had not asked to see their MCH booklet (Table 2). Similarly, 36% of caregivers (194/541) felt that they did not know or were unsure of the vaccines their child needs (Table 2). Of the children who were vaccinated on the day of the survey, half of their caregivers (85/165) reported that they were informed what vaccines their children were given and one-third (56/184) reported that they were informed about potential adverse events following immunization. Almost all caregivers reported that they had never been asked to pay for a vaccine (95%; 523/551)); 17% (92/549) of the caregivers reported that they had been asked to pay for a MCH booklet at some time in the past.

**Missed opportunities for vaccination.** *Child-based MOV prevalence*. Out of the 558 children with documented vaccination dates, 324 were eligible for at least one vaccine dose during the visit on the survey date (Fig 1, Table 3). Of these 324 children, 25% (80) of the children were vaccinated with all the eligible vaccine doses, 18% (58) received some, but not all of the vaccine doses that they were eligible for, and 57% (186) were not given any of their eligible doses during the visit. Overall, 244 children who were determined to be eligible to receive due or delayed vaccine doses remained unvaccinated after the health service encounter, resulting in a child-based MOV prevalence of 75% in this study sample.

When stratified by reason for visit, 93% (158/169) of children who visited the health facility for reasons other than vaccination (e.g. medical consultation, healthy child visit, accompanying an adult, hospitalization, etc.) with at least one eligible dose due remained incompletely vaccinated at the end of the health service encounter. Among those children visiting *specifically* for vaccination with at least one eligible dose due, 55% (86/155) had a MOV. By age categories, children who were 12 months or older had higher prevalence of MOV (90%; 61/68) than children who were less than 12 months (71%; 183/256).

**Timeliness with reference to the nationally recommended schedules.** Birth dose vaccines had the highest proportion of doses given in a timely manner, with bacille Calmette-

**Table 2. Characteristics of surveyed caregivers of children with documented vaccination dates, missed opportunities for vaccination (MOV) assessment, Kenya, 2016.**

|  | n | % |
|---|---|---|
|  | **558** |  |
| *Child demographics* |  |  |
| **Sex** | **550** |  |
| Male | 286 | 52 |
| Female | 264 | 48 |
| **Age** | **558** |  |
| <12 months | 420 | 75 |
| ≥12 months | 138 | 25 |
| **Ever vaccinated** | **551** |  |
| Yes | 542 | 98 |
| No | 9 | 2 |
| *Caregiver demographics* |  |  |
| **Relationship to child** | **554** |  |
| Mother | 527 | 95 |
| Father | 4 | 1 |
| Other (Grandparent, Uncle/Aunt, Sibling, Other) | 23 | 4 |
| **Caregiver can read and write** | **462** |  |
| Yes | 406 | 88 |
| No | 56 | 12 |
| **Educational Level** | **554** |  |
| None | 42 | 8 |
| At least some primary | 308 | 56 |
| At least some secondary | 204 | 37 |
| *Health facility visit* |  |  |
| **What was your reason for visiting the health facility today?** | **558** |  |
| Medical consultation | 149 | 27 |
| Vaccination | 182 | 33 |
| Healthy child visit or check-up | 154 | 28 |
| Child is accompanying adult or sibling | 50 | 9 |
| Hospitalization | 2 | 0 |
| Other or no reason reported | 21 | 4 |
| **Does your child have a mother-and-child health (MCH) booklet?** | **556** |  |
| Yes, and it is available at this visit | 509 | 92 |
| Yes, but not available at today's visit | 38 | 7 |
| No | 9 | 2 |
| **Do you know the vaccines your children need and when given?** | **541** |  |
| Yes | 347 | 64 |
| No | 107 | 20 |
| Not sure | 87 | 16 |
| **Did staff ask for the MCH booklet?** | **555** |  |
| Yes | 426 | 77 |
| No | 129 | 23 |
| **Have you ever been asked to pay for a vaccine?** | **551** |  |
| Yes | 28 | 5 |
| No | 523 | 95 |
| **Have you ever been asked to pay for a MCH booklet?** | **549** |  |

*(Continued)*

**Table 2.** (Continued)

|  | n | % |
|---|---|---|
| Yes | 92 | 17 |
| No | 457 | 83 |
| **Was the child vaccinated here today?** | **551** |  |
| Yes | 184 | 33 |
| No | 367 | 67 |
| **Were you told what vaccines your child was given[1]** | **165** |  |
| Yes | 85 | 52 |
| No | 80 | 48 |
| **Were you told about potential adverse reactions following immunization?[1]** | **184** |  |
| Yes | 56 | 30 |
| No | 128 | 70 |
| **Were you satisfied with the service you received today?** | **186** |  |
| Yes | 167 | 90 |
| No | 19 | 10 |

[1]Asked only of caregivers who indicated the child was vaccinated at the health facility on day of the survey (n = 184). Unless otherwise noted, questions were asked of all caregivers.

Guerin (BCG) vaccine and oral polio vaccine (OPV) birth dose at 81% and 83%, respectively (Table 4). For multi-dose vaccine series, timeliness generally decreased with later doses. Similarly, timeliness of MCV, due to be given at 9 and 18 months, dropped nine percentage points from the first dose (75%; 127/169) to the second dose (66%; 21/32).

## Health worker KAP survey

**Demographics of surveyed health workers.** KAP surveys were completed by 376 health workers, of whom 60% were female and 65% were under the age of 30 (Table 5). Half (50%) were nurses or midwives and 62% had less than five years of clinical experience.

**Health worker knowledge, attitudes, practices.** Less than half (41%) of health workers reported ever receiving training on vaccination or vaccine-preventable diseases (Table 5). However, most health workers were able to identify vaccines that children should routinely receive (88–97%), including BCG, pentavalent, MCV, polio, rotavirus, and PCV. However, about one in five (19%) could not correctly identify *all* the routine vaccines. In addition, one in four of those surveyed (25%) made mistakes in describing the vaccination schedule for BCG, OPV, pentavalent vaccine, and MCV first dose.

Among health workers who reported administering vaccines as part of their daily work routine, 39% (55/142) reported that they did not have the materials they needed for patients seeking immunization services (Table 5). Among those reporting missing materials, 32% (15/47) were missing vaccines, 34% (16/47) were missing syringes, and 68% (32/47) said they were missing MCH booklets. When giving a new MCH booklet to caregivers, 66% (93/141) of health workers indicated that they usually instruct caregivers to keep the booklet safe and bring the booklet to all health facility visits.

## Discussion

The MOV assessment in Kenya found a high prevalence of MOV (75%) among children <24 months of age visiting selected health facilities. MOV rates were very high among those visiting for non-vaccination-related reasons with only a small percentage receiving any vaccines. In

**Table 3. Prevalence of missed opportunities for vaccination (MOV)[1] among surveyed children, by reason for visit and age, missed opportunities for vaccination (MOV) assessment, Kenya, 2016.**

| KENYA | Total children with documented vaccination dates | On arrival for the health visit: Number of children needing 1+ eligible due doses | During the health visit: All eligible doses given | | During the health visit: Some eligible doses given (not all) | | During the health visit: No eligible doses given | |
|---|---|---|---|---|---|---|---|---|
| | | | n | % | n | % | n | % |
| **Age** | n | n | n | % | n | % | n | % |
| <12 months | 420 | 256 | 73 | 29 | 55 | 21 | 128 | 50 |
| ≥12 months | 138 | 68 | 7 | 10 | 3 | 4 | 58 | 85 |
| *Total* | 558 | 324 | 80 | 25 | 58 | 18 | 186 | 57 |
| **Reason for visit** | n | n | n | | n | | n | |
| Vaccination | 182 | 155 | 69 | 45 | 55 | 35 | 31 | 20 |
| *Non-vaccination visit* | | | | | | | | |
| Medical consultation | 149 | 83 | 0 | 0 | 2 | 2 | 81 | 98 |
| Healthy child visit or check-up | 154 | 51 | 6 | 12 | 1 | 2 | 44 | 86 |
| Child is accompanying adult | 50 | 23 | 0 | 0 | 0 | 0 | 23 | 100 |
| Hospitalization | 2 | 1 | 0 | 0 | 0 | 0 | 1 | 100 |
| Other | 11 | 8 | 5 | 63 | 0 | 0 | 3 | 38 |
| No reason reported | 10 | 3 | 0 | 0 | 0 | 0 | 3 | 100 |
| *Non-vaccination visit total* | 376 | 169 | 11 | 7 | 3 | 2 | 155 | 92 |
| *Total* | 558 | 324 | 80 | 25 | 58 | 18 | 186 | 57 |
| | | *Total MOV (some, but not all eligible doses given or no eligible doses given)[2]* | | | | | 244 | **75** |

[1]Missed opportunity for vaccination (MOV): contact with health services by a child (or adult) who is eligible for vaccination (unvaccinated, partially vaccinated/not up-to-date, and free of contraindications to vaccination), which does not result in the individual receiving all the vaccine doses for which he or she is eligible) [1, 2].

[2]Among the subset of children with documented vaccination dates and eligible for one or more vaccine doses (n = 324).

addition, approximately half of children coming for vaccination visits did not receive all vaccines for which they were eligible. The health worker survey revealed inadequate knowledge and poor practices, as well as a reported lack of resources needed to vaccinate all eligible children visiting health facilities in Kenya. Nearly one-quarter of all caregivers of children visiting for all reasons reported that the health worker had not asked for their child's vaccination record (MCH booklet) at the time of the visit. These findings have implications for the national program in Kenya. Routine vaccination checks by health workers during all health facility visits, including non-vaccination visits, has the potential for increasing coverage and equity in Kenya. In addition, processes to ensure that adequate vaccination-related supplies are available at facilities offering immunization services during all visit types and for all age groups will be necessary.

The findings from the qualitative component of the assessment and other studies also highlight inconsistent vaccination checks, particularly during non-vaccination visits, or among certain populations [3, 4, 28]. In order to ensure children are not missed, efforts should be made to institute routine checking of children's MCH booklet as a standard practice, particularly at non-vaccination visits and among children ≥12 months old. To maximize their benefits, MCH booklets must be complete and legible [7]. Unfortunately, when health workers are faced with competing tasks, completing vaccination records can be among the first activities to be de-prioritized [7, 33].

To be effective, vaccination record checks require that health workers possess adequate knowledge of the antigens in the national schedule, are able to assess eligibility by age based on

**Table 4. Timeliness of vaccine doses administered to surveyed children with documented vaccination histories, missed opportunities for vaccination (MOV) assessment, Kenya, 2016.**

| Vaccine dose | Total number of children who received dose[2] | Timeliness[1] | | |
|---|---|---|---|---|
| | | Too early % | Timely % | Late % |
| **At birth** | | | | |
| BCG[3] | 513 | -- | 81 | 19 |
| OPV[4] | 443 | -- | 83 | 17 |
| **First dose** | | | | |
| OPV[3] | 482 | 15 | 68 | 17 |
| Pentavalent[5] vaccine | 497 | 13 | 70 | 18 |
| PCV[6] | 484 | 13 | 69 | 18 |
| Rotavirus vaccine | 471 | 13 | 69 | 18 |
| **Second dose** | | | | |
| OPV[3] | 425 | 8 | 70 | 22 |
| Pentavalent[5] vaccine | 442 | 8 | 70 | 22 |
| PCV[6] | 428 | 8 | 70 | 22 |
| Rotavirus vaccine | 409 | 7 | 69 | 24 |
| **Third dose** | | | | |
| OPV[3] | 375 | 6 | 64 | 30 |
| Pentavalent[5] vaccine | 389 | 5 | 64 | 30 |
| PCV[6] | 369 | 6 | 63 | 31 |
| **MCV[7] first dose** | 169 | 19 | 75 | 6 |
| **MCV[7] second dose** | 32 | 34 | 66 | 0 |

[1] Please see Table 1 for intervals and immunization schedule used for this analysis.

[2] Children with documented history of receiving a dose either on the day of survey or previously.

[3] bacille Calmette-Guerin (BCG) vaccine.

[4] Oral poliovirus vaccine (OPV).

[5] Diphtheria-pertussis-tetanus-hepatitis B-Haemophilus influenzae type b (Pentavalent).

[6] Pneumococcal conjugate vaccine (PCV).

[7] Measles containing vaccine (MCV).

the schedule, and have access to the necessary job aids they need to support them in this. In our study, approximately one in five surveyed health workers were unable to identify all the vaccines in the national schedule and one in four were unable to identify the correct schedule for BCG, OPV, pentavalent vaccine, and MCV first dose. Overall, all health workers must be trained on the vaccination schedule and further work may be useful to identify exact gaps in knowledge within the immunization schedule for targeted education. Additionally, although the questionnaire did not test their understanding of catch-up schedules, previous studies show this to be a confusing concept for many health workers and it is necessary to also ensure health workers understand the national policy on catch-up vaccination for children with a delayed schedule [5, 29]. As Kenya has continued to add more antigens to their national immunization schedule, with varying target age groups (MCV second dose in 2015, IPV and YF in 2016 and human papillomavirus vaccine in 2019), the potential for MOV has increased and will accelerate further. It is important to ensure that all health workers who interface with patients, regardless of whether or not they work in immunizations, are equipped with the knowledge, job aids, and support to handle the multiple scenarios that children with delayed and out-of-sync schedules may present.

Finally, health workers cannot deliver vaccines to children when the vaccines or related materials are out of stock. Two out of five surveyed health workers reported that they did not

**Table 5. Characteristics and knowledge, attitudes, and practices of surveyed health workers, missed opportunities for vaccination (MOV) assessment, Kenya, 2016.**

| | n | % |
|---|---|---|
| | 376 | |
| *Health worker demographics* | | |
| **Sex** | 371 | |
| Male | 147 | 40 |
| Female | 224 | 60 |
| **Age** | 373 | |
| <19 | 127 | 34 |
| 20–29 | 115 | 31 |
| 30–39 | 81 | 22 |
| 40–49+ | 50 | 13 |
| **What is your professional training?** | 374 | |
| Doctor | 7 | 2 |
| Nurse/Midwife | 188 | 50 |
| Clinical Officer | 59 | 16 |
| Public Health Officer | 14 | 4 |
| Lab or pharmaceutical technologist | 40 | 11 |
| Health or information records officer | 11 | 3 |
| Nutritionist | 13 | 3 |
| Pharmacist | 9 | 2 |
| Other | 33 | 9 |
| **Number of years of clinical experience** | 374 | |
| 0–4 years | 232 | 62 |
| 5–9 years | 85 | 23 |
| 10+ | 57 | 15 |
| **Have you ever been trained in vaccination or vaccine-preventable diseases?** | 373 | |
| Yes | 154 | 41 |
| No | 219 | 59 |
| **When were you last trained?** | 153 | |
| <1 year ago | 41 | 27 |
| 1–2 years ago | 15 | 10 |
| 2–3 years ago | 69 | 45 |
| >4 years ago | 28 | 18 |
| *Health worker knowledge, attitudes, practices* | | |
| **Which vaccines should healthy children receive?[1]** | 376 | |
| BCG[2] | 365 | 97 |
| MCV[3] | 369 | 98 |
| Pentavalent[4] | 344 | 91 |
| Polio vaccine | 366 | 97 |
| Rotavirus vaccine | 328 | 87 |
| PCV[5] | 329 | 88 |
| Selected all of the above | 304 | 81 |
| **Could correctly identify schedule for BCG[2], oral polio vaccine, Pentavalent[4], and MCV first dose** | 291 | |
| Yes | 217 | 75 |
| No | 74 | 25 |
| **What are the contraindications for any vaccine[1]** | 352 | |
| Local reaction to previous dose | 93 | 26 |

*(Continued)*

**Table 5.** (Continued)

| | n | % |
|---|---|---|
| Low-grade fever | 43 | 12 |
| Seizures under medical treatment | 81 | 23 |
| Pneumonia and other serious diseases | 74 | 21 |
| None of the above | 144 | 41 |
| **When should vaccination status be assessed?** | 373 | |
| Child's wellness/routine visit | 53 | 14 |
| Consultation for any illness | 24 | 6 |
| When a child is accompanying a woman visiting a healthcare facility for any reason | 35 | 9 |
| All of the above | 261 | 70 |
| **Who should evaluate children's vaccination status?** | 376 | |
| The child's parents | 13 | 3 |
| The nurse responsible for immunization | 124 | 33 |
| Physicians in external consultations, inpatient services, and emergency rooms | 7 | 2 |
| All of the above | 232 | 62 |
| **Do you administer vaccines as part of your routine job?** | 376 | |
| Yes | 142 | 38 |
| No | 234 | 62 |
| **What instructions do you give caregivers when you give them a new mother-and-child health (MCH) booklet?[1,6]** | 141 | |
| Keep this booklet safe (only) | 105 | 74 |
| Bring this booklet to all visits to the health facility (only) | 134 | 95 |
| Keep this booklet safe *and* bring this booklet to all visits to the health facility | 93 | 66 |
| Bring this booklet only when you come for vaccinations | 7 | 5 |
| Other | 3 | 2 |
| **Today, I have enough materials for the patients seeking immunization services[6]** | 142 | |
| Agree | 87 | 61 |
| Disagree | 55 | 39 |
| **What is missing?[7]** | 47 | |
| Vaccines | 15 | 32 |
| Syringes | 16 | 34 |
| Recording materials | 7 | 15 |
| MCH booklets | 32 | 68 |
| Other | 4 | 9 |

[1] Respondents were allowed to select multiple responses.

[2] measles containing vaccine (MCV).

[3] bacille Calmette-Guerin (BCG) vaccine.

[4] Diphtheria-pertussis-tetanus-hepatitis B-Haemophilus influenzae type b (Pentavalent) vaccine.

[5] Pneumococcal conjugate vaccine (PCV).

[6] Question asked to health workers who indicated that they administer vaccines as part of their routine job (n = 142).

[7] Among those who disagreed that there were enough materials for the patients seeking immunization services.

have enough materials for children seeking immunization, with vaccines, syringes, and MCH booklets most often identified as missing. The consequence is that eligible children remain unvaccinated and susceptible to vaccine-preventable diseases following a health facility visit. When MCH booklets are out of stock, missing records may make it difficult for health workers to track the child's vaccination status in the future. MCH booklets were frequently cited by health workers as a missing item and from the qualitative assessment, we learned the

importance of these MCH booklets to caregivers, giving them ownership over their child's health [28]. Inadequate stocks create barriers to caregivers seeking immunization [3, 34]. Efforts must be made to ensure that health facilities are equipped with adequate stocks of vaccine doses and vaccination-related supplies needed for all children seeking immunization and other health services.

Following discussion of the findings from this MOV assessment, the multi-partner technical working group on immunization in Kenya endorsed an action plan to reduce MOV. Kenya is standardizing health worker practices by disseminating updated Kenyan NVIP manuals and standard operating procedures. The MoH has also implemented an orientation package, aimed specifically at non-NVIP health staff, to improve health worker knowledge on immunization and practices across all departments. This orientation package includes training modules on vaccination practices and interpersonal communication skills utilizing adult learning strategies. To reduce the likelihood of stock-outs of vaccines, a stock management module is being implemented across all counties. The national level is also distributing electronic copies of vaccination recording materials, including monitoring charts, summary sheets, tally sheets, and MCH booklets, to allow them to be printed at the county level as well as by private health facilities for easier access.

## Limitations

Due to the sampling methodology of MOV assessments, this assessment was not nationally representative and the results cannot be interpreted to represent MOV rates across all Kenya health facilities. Additionally, because this was a health facility-based assessment, caregivers that visited the health facility on the day of the assessment may differ from others in the community. Next, although all questionnaires were piloted and adapted to the country-context prior to the assessment, there were still areas in which they could have been improved; certain modifications may have improved the quality of the survey by ensuring clarity of questions, responses, and appropriate skip patterns. Similarly, questionnaires were only available in English, but were sometimes administered in Swahili. This may have resulted in varied translations causing differing understandings of questions.

Finally, the estimation of MOV was limited to children with *documented* vaccination dates; verbal recall was not accepted. If children without documented vaccination dates are more likely to have a MOV, the true prevalence of MOV in Kenya is likely to be higher than we have reported.

## Conclusion

The MOV assessment conducted in Kenya proved to be a low-resource approach that identified easily-implementable but potentially very impactful activities to improve vaccination coverage. Kenya's intervention plan to address MOV must continue to be scaled up across the country in order to reduce MOV, increase routine immunization, reduce outbreaks of vaccine-preventable diseases and further reduce infant mortality.

## Supporting information

**S1 File. Clarification of ethical review for standard program reviews in Kenya.**
(PDF)

**S2 File. Missed opportunities for vaccination assessment exit interview survey: Kenya, 2016.**
(PDF)

**S3 File. Missed opportunities for vaccination assessment health worker survey: Kenya, 2016.**
(PDF)

## Acknowledgments

The authors thank the caregivers, health workers, and healthcare administrators who gave of their time to participate in the Kenya MOV assessment. They also acknowledge the assistance of the entire Kenya *MOV Team*, Kenya Ministry of Health, the country offices of the World Health Organization, John Snow, Inc. (JSI) Maternal and Child Survival Program (MCSP), the Clinton Health Access Initiative (CHAI), Health NGOs Network (HENNET), UNICEF, the Inter-religious council of Kenya, the American Red Cross, and other local immunization partners during the assessment and subsequent implementation of interventions to reduce missed opportunities for vaccination in Kenya.

## Disclaimer

The authors alone are responsible for the views expressed in this article, which do not necessarily represent the views, decisions, or policies of the institutions with which the authors are affiliated.

## Author Contributions

**Conceptualization:** Collins Tabu, Ikechukwu Udo Ogbuanu.

**Data curation:** Anyie J. Li, Ikechukwu Udo Ogbuanu.

**Formal analysis:** Colin Sanderson.

**Investigation:** Anyie J. Li, Collins Tabu, Stephanie Shendale, Kibet Sergon, Peter O. Okoth, Isaac K. Mugoya, Zorodzai Machekanyanga, Iheoma U. Onuekwusi, Ikechukwu Udo Ogbuanu.

**Methodology:** Collins Tabu, Kibet Sergon, Iheoma U. Onuekwusi, Ikechukwu Udo Ogbuanu.

**Project administration:** Collins Tabu, Kibet Sergon, Ikechukwu Udo Ogbuanu.

**Resources:** Collins Tabu, Stephanie Shendale, Kibet Sergon, Peter O. Okoth, Iheoma U. Onuekwusi, Ikechukwu Udo Ogbuanu.

**Supervision:** Anyie J. Li, Collins Tabu, Stephanie Shendale, Kibet Sergon, Peter O. Okoth, Isaac K. Mugoya, Zorodzai Machekanyanga, Iheoma U. Onuekwusi, Ikechukwu Udo Ogbuanu.

**Validation:** Colin Sanderson, Ikechukwu Udo Ogbuanu.

**Writing – original draft:** Anyie J. Li.

**Writing – review & editing:** Collins Tabu, Stephanie Shendale, Kibet Sergon, Peter O. Okoth, Isaac K. Mugoya, Zorodzai Machekanyanga, Iheoma U. Onuekwusi, Colin Sanderson, Ikechukwu Udo Ogbuanu.

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
