## [Decision Letter · Decision Letter 0]

15 Jun 2020

PONE-D-19-31527

Assessment of missed opportunities for vaccination in Kenyan health facilities, 2016

PLOS ONE

Dear Dr Anyie Joana Li

Thank you for submitting your manuscript to PLOS ONE. After careful consideration, we feel that it has merit but does not fully meet PLOS ONE’s publication criteria as it currently stands. Therefore, we invite you to submit a revised version of the manuscript that addresses the points raised during the review process.

We would appreciate receiving your revised manuscript by June 10. To enhance the reproducibility of your results, we recommend that if applicable you deposit your laboratory protocols in protocols.io, where a protocol can be assigned its own identifier (DOI) such that it can be cited independently in the future. For instructions see: http://journals.plos.org/plosone/s/submission-guidelines#loc-laboratory-protocols

We look forward to receiving your revised manuscript.

Kind regards,

Daniela Flavia Hozbor

Academic Editor

PLOS ONE

Journal Requirements:

2. Please provide additional details regarding participant consent.

In the ethics statement in the Methods and online submission information, please ensure that you have specified whether consent was written or verbal/oral.

If consent was verbal/oral, please specify: a) whether the ethics committee approved the verbal/oral consent procedure, b) why written consent could not be obtained, and c) how verbal/oral consent was recorded.

Since your study included minors under age 18, please state whether you obtained consent from parents or guardians in these cases.

Please also indicate in your Ethics Statement whether all data were anonymised before the study authors accessed them.

3. Please note that all PLOS journals ask authors to adhere to our policies for sharing of data and materials: https://journals.plos.org/plosone/s/data-availability. According to PLOS ONE’s Data Availability policy, we require that the minimal dataset underlying results reported in the submission must be made immediately and freely available at the time of publication. As such, please remove any instances of 'unpublished data' or 'data not shown' in your manuscript and replace these with either the relevant data (in the form of additional figures, tables or descriptive text, as appropriate), a citation to where the data can be found, or remove altogether any statements supported by data not presented in the manuscript.

Reviewers' comments:

Reviewer's Responses to Questions

**Comments to the Author**

1. Is the manuscript technically sound, and do the data support the conclusions?

Reviewer #1: Yes

Reviewer #2: Yes

2. Has the statistical analysis been performed appropriately and rigorously? 

Reviewer #1: Yes

Reviewer #2: Yes

3. Have the authors made all data underlying the findings in their manuscript fully available?

Reviewer #1: Yes

Reviewer #2: Yes

4. Is the manuscript presented in an intelligible fashion and written in standard English?

Reviewer #1: Yes

Reviewer #2: Yes

5. Review Comments to the Author

Reviewer #1: The introduction and discussion can use a better framing of the purpose and implication of quantifying the missed opportunities. It seems a bit obvious that all contacts that did not result in vaccination if eligible will be a missed opportunity. Similarly, if health workers were not instructed to provide vaccination at every contacts that were not primarily for reasons of vaccinations, it is strange to ding them for not doing so.

Also it will help to give some context to the quantified missed opportunities. How does this compare to other countries? How bad is this? There was a quick description of the missed opportunities being much more if taking into account those without documentation of vaccination dates, but more can be said to further provide context to help readers understand the gravity of your results

Reviewer #2: The authors have provided a good documentation of findings regarding missed opportunities for simultaneous vaccines. The methods for data collection and analysis are described reasonably, such that readers can readily understand the basis and organization of results, without delving unnecessarily deeply into statistical methods. As such this study provides a good foundational work of reporting as a benchmark for future studies. To improve this study the authors might consider discussing the results in the context of specific action, rather than as a more general reporting of data. For example, the authors state that one in five surveyed health workers were unable to identify all of the vaccines in the national schedule. It could be additionally useful to know if there were specific vaccines or points in the recommended vaccination schedule that could be a point of focus for improvement. Similarly, cross-tabulation of demographic variables against timeliness and MOV data cold provide additional insight into specific characteristics that tend to favor missed opportunities or late vaccines - or indeed those characteristics that favor no missed opportunities and consistently timely vaccines. And, adding a statistical measure of significance, a "p" value, would provide to the reader a distinction between general reporting and significant findings.

6. PLOS authors have the option to publish the peer review history of their article (what does this mean?). If published, this will include your full peer review and any attached files.

Reviewer #1: Yes: Yvonne Tam

Reviewer #2: No

---

## [Author Response · Author response to Decision Letter 0]

3 Aug 2020

Thank you for your review. Please see the attached 'Response to Reviewers' for responses to each comment.

---

## [Editor Report · Decision Letter 1]

6 Aug 2020

Assessment of missed opportunities for vaccination in Kenyan health facilities, 2016

PONE-D-19-31527R1

Dear Dr. Anyie Joana Li,

We’re pleased to inform you that your manuscript has been judged scientifically suitable for publication and will be formally accepted for publication once it meets all outstanding technical requirements.

Kind regards,

Daniela Flavia Hozbor

Academic Editor

PLOS ONE
---

## [Editor Report · Acceptance letter]

10 Aug 2020

PONE-D-19-31527R1 

Assessment of missed opportunities for vaccination in Kenyan health facilities, 2016

Dear Dr. Li:

I'm pleased to inform you that your manuscript has been deemed suitable for publication in PLOS ONE. Congratulations! Your manuscript is now with our production department. 

Kind regards, 

on behalf of

Dr. Daniela Flavia Hozbor 

Academic Editor

PLOS ONE